# VisCon-100K: Leveraging Contextual Web Data for Fine-tuning Vision Language Models with Leaky Visual Conversations

## Abstract

Vision-language models (VLMs) excel in various visual benchmarks but are often constrained by the lack of high-quality visual fine-tuning data. To address this challenge, we introduce VisCon-100K, a novel dataset derived from interleaved image-text web documents. Our approach transforms 45K web documents from the OBELICS dataset into 100K image conversation samples. We utilize GPT-4V to generate image-contextual captions and OpenChat 3.5 model to convert these captions into diverse free-form and multiple-choice question-answer pairs. Integrating this dataset for fine-tuning considerably enhances VLM performance across multiple benchmarks. Unlike methods that focus solely on fine-grained visual content, our approach leverages accompanying web context, yielding superior results. We also discover that a 'leaky modality mix,' where conversation samples contain questions answerable from both the image and its contextual caption, outperforms non-leaky combinations of captions and Q&A pairs. Our dataset shows strong performance with two popular VLM approaches: text-only large language model (LLM) aligned with a vision encoder using image captions data (ShareGPT4V-7b) and multimodally pretrained LLM (IDEFICS2-8b) using interleaved image-text data. In addition to releasing the VisCon-100K dataset, we provide a contextual captioner trained on this dataset, facilitating scalable fine-tuning data generation for future research and open-source applications.

## 1 Introduction

Recent advancements in large language models (LLMs) have revolutionized natural language processing (NLP), significantly impacting tasks such as text generation, summarization, translation, and question-answering. Models like LLaMA-2 (Touvron et al., 2023) and Mistral (Jiang et al., 2023) have demonstrated exceptional capabilities, driving extensive research into their applications across various domains. Inspired by these successes, researchers have explored adapting LLMs for visual tasks, leading to significant developments in vision-language models (VLMs).

Two primary approaches have emerged for integrating visual understanding into LLMs:

1. **Alignment using Image Captions**: Popular models such as LLaVA-1.5 (Liu et al., 2024a) and ShareGPT4V (Chen et al., 2023) combine a pre-trained LLM with a CLIP (Radford et al., 2021)-based image encoder. The alignment of the image encoder's output with the LLM is achieved through a two-stage training process: initially aligning the two modalities using image captions, followed by fine-tuning on vision-language tasks such as visual question answering (VQA).

2. **Multimodal Pretraining using Interleaved Image-Text**: These methods, including Kosmos-1 (Huang et al., 2024) and IDEFICS2 (Laurençon et al., 2024b), adopt a different strategy by performing multimodal pretraining. Using interleaved image-text web documents, they perform textual next-token prediction while incorporating visual context. This is typically followed by fine-tuning with VQA datasets.

In addition to these two dominant approaches, several other methods such as Flamingo (Alayrac et al., 2022), MiniGPT-4 (Zhu et al., 2023), Prismer (Liu et al., 2023a), Chameleon (Lu et al.,

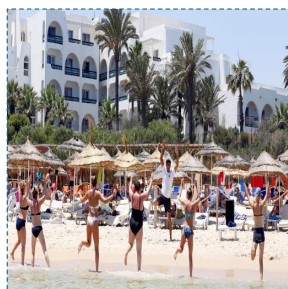

**Webpage Context:** <another-image>
New pictures show holidaymakers lying on the beach sunbathing in Sousse, the region where 38 people were shot dead by an ISIS terrorist in 2015.
<another-image> <another-image>
They had abandoned the country's paradise beaches after the attacks in Port El Kantaoui, where 30 of the dead were British.
In one photo a British girl is seen sunbathing in a Union Jack bikini.
Tourists from Russia and Algeria have also returned to Tunisia for their holidays and the new photos show that Sousse is now thriving again after three years of near-empty beaches.
<another-image> <another-image> <another-image>
But it's not all fun and games, with the presence of the Tunisian National Guard being felt as a boat patrols the shores outside the Jaz Tour Khalef Hotel in Sousse. …
With tourists returning to Tunisia, counter-terrorism chiefs have put out a video advising people what to do if they are caught up in a terrorist attack overseas.
The four-minute film adapts the Run, Hide, Tell message and depicts a firearms attack at a hotel.
<image> <another-image> <another-image>
The attack was the deadliest non-state attack in the history of modern Tunisia and saw 30 Brits killed in a bloody rampage which the resort owners 'couldn't have stopped'. …

**Contextual Caption:** The image depicts a lively beach scene in a resort area, specifically in Sousse, Tunisia. The photo captures a moment of joy and activity where a group of tourists is participating in what appears to be an aqua aerobics class directly on the sandy shore. In the foreground, several women can be seen with their arms extended, simultaneously jumping and kicking up water, suggesting a choreographed fitness-related group activity. Their facial expressions and body language convey energy and enthusiasm. In the background, other holidaymakers are seen engaging in more leisurely beach activities; some are lounging under straw umbrellas on sunbeds while others walk along the beach. The architecture of a white Mediterranean-style hotel overlooks the scene, indicative of the resort's accommodation offerings. This image underlines the resurgence of tourism in the area, highlighting the return of visitors after a period of absence due to security concerns following the tragic events of 2015. The scene conveys a sense of normalcy and enjoyment, with tourists once again indulging in the beach experiences that the destination is known for. The energy and excitement captured in the photo align with the article's broader narrative, indicating that Sousse's beaches have regained their status as an attractive destination for international visitors, including those from Britain, after the lifting of travel restrictions. The image stands in contrast to the somber history of the site, showing a moment of recovery and recreation.

**Non-Contextual Caption:** This image depicts a lively beach scene in front of a resort. A group of individuals appears to be enjoying themselves on the sandy shore, participating in what looks like a spontaneous group dance or an action shot intended for a photograph. There are seven people in the focus of the image, spread out along the water's edge. They are all in motion, likely jumping, and have their arms extended, which suggests they are mid-dance or jump. Each person exhibits a different stage of movement, indicating dynamic and playful energy. The group is composed of both males and females, wearing summer beachwear such as shorts and swimsuits, embodying the leisurely atmosphere of the beach. In the background, there is a large white resort building, several floors high, with balconies, archways, and a variety of windows. A neat row of straw umbrellas and beach loungers under them suggests this is a well-maintained tourist area. Many loungers are occupied by beachgoers enjoying the sun. The vegetation present includes palm trees that rise above the resort, conveying a tropical or Mediterranean climate. The clear blue sky is nearly cloudless, implying it is a sunny and favourable day for beach activities. The overall impression is that of a joyful, energetic moment shared among friends or family, epitomizing the carefree nature of a holiday by the sea.

Figure 1: An OBELICS web document with generated contextual and non-contextual captions. The non-contextual caption describes the image in isolation, while the contextual caption integrates additional information from the surrounding web text, highlighted in red, providing a more nuanced and comprehensive description.

2024), and Meta-Transformer (Zhang et al., 2023) adapt text-only LLMs for visual tasks. However, these alternatives, often involving more complex techniques, generally underperform on similar data and compute budgets compared to ShareGPT4V and IDEFICS2. Also, fine-tuning VLMs require considerable computational resources. So we evaluate our dataset and its design with the top representative models across two different popular VLM approaches: text-only large language models (LLM) aligned with a vision encoder using image captions data (ShareGPT4V-7b) and multimodally pretrained LLM (IDEFICS2-8b) using interleaved image-text data.

Despite these advancements, a critical gap persists: the scarcity of high-quality, diverse visual fine-tuning datasets. While extensive text-only fine-tuning datasets exist (Liu et al., 2024c), there is a notable lack of vision-language datasets (Laurençon et al., 2024b) that provide the contextual richness required for effective vision-language understanding. Current datasets often fall short in capturing the broader web-based context that can enhance vision-language understanding.

To bridge this gap, we introduce **VisCon-100K**, a contextually rich dataset derived from interleaved image-text web documents. Our pipeline processes 45K web documents from the OBELICS (Laurençon et al., 2024a) dataset into 100K image conversation samples. These samples are created by generating image-contextual captions using OpenAI GPT-4V API and transforming them into diverse free-form and multiple-choice question-answer pairs using OpenChat 3.5 (Wang et al., 2023). The resulting dataset, VisCon-100K, captures both **fine-grained visual descriptions** and **broader contextual information**, enabling more effective fine-tuning of VLMs.

Our contributions can be summarized as follows:

1. **Effective Use of Contextual Web Data**: We demonstrate the effectiveness of using contextual web data in combination with images, showcasing a sophisticated data generation pipeline that can be extended for future research and applications.

2. **VisCon-100K Dataset**: We provide a novel, scalable dataset that notably enhances the performance of vision-language models across multiple benchmarks. By leveraging web context, VisCon-100K offers a richer and more diverse training resource than existing datasets.

3. **Contextual Captioner**: We provide a trained contextual captioner to support scalable fine-tuning, enabling further research and open-source applications by generating high-quality contextual captions without relying on paid services like GPT-4V.

4. **Leaky Modality Mix**: We introduce the concept of a "leaky modality mix," where conversation samples contain questions that can be answered from both the image and its contextual caption. This mix facilitates better integration of visual and textual information, outperforming non-leaky combinations of captions and Q&A pairs.

By addressing the need for high-quality visual fine-tuning data and demonstrating the benefits of incorporating contextual information, VisCon-100K represents a major step forward in the development of robust vision-language models.

## 2 RELATED WORK

Creating high-quality datasets for fine-tuning vision-language models is essential for improving their performance on complex multimodal tasks. Existing methods have made significant strides in this area, yet various challenges persist in terms of diversity, contextual richness, and scalability. Here, we discuss notable contributions and their limitations, setting the stage for the introduction of our approach used to develop **VisCon-100K**.

**Vision-Language Dataset Creation**

1. **Fine-Grained Image Captions**: Approaches such as those used in **ShareGPT4V** (Chen et al., 2023), **FuseCap** (Rotstein et al., 2023), and **GranD** (Rasheed et al., 2024) generate detailed image descriptions using LLMs. ShareGPT4V employs the GPT-4V API to produce detailed seed captions, aiming to reduce hallucinations and enhance dataset quality. Similarly, FuseCap integrates visual information from sources like object detectors and image taggers to enrich the captions, while GranD also queries LLM with a scene graph to add extra context. However, as these datasets scale, they tend to produce redundant descriptions of similar visual content, limiting their diversity and informativeness.

2. **Contextual Data Utilization**: Some models, like **IDEFICS-2** (Laurençon et al., 2024b) and **Flamingo** (Alayrac et al., 2022), employ contextual data in their pretraining by using interleaved image-text web documents. However, these approaches often retain a weak dependency on images while focusing on textual next-token prediction. The lack of grounding in the visual content means that the context derived from the web documents does not fully integrate with the image data, resulting in suboptimal alignment between visual and textual modalities.

3. **Repurposing Classical Computer Vision Datasets**: Other methods, like **LLaVA** (Liu et al., 2024b), **ALLaVA** (Chen et al., 2024) and **IDEFICS-2** (Laurençon et al., 2024b), attempt to repurpose datasets from common computer vision tasks for vision-language fine-tuning. While useful, these datasets often lack the diversity and contextual richness needed for real-life image conversations. They typically provide limited contextual information and fail to capture the broader web-based context that can enhance vision-language understanding. Moreover, these datasets often exhibit modality isolation, where questions are answerable either from a visual or a textual modality, but not both.

**Challenges and Limitations**

- **Redundancy**: A common issue with current methods is the generation of redundant information, especially when scaling up the dataset. Repeated descriptions of similar content can reduce the dataset's overall effectiveness in training robust VLMs.

- **Lack of Contextual Grounding**: Many approaches show limited ability to generate data that is both contextually rich and relevant to real-life applications.

- **Modality Isolation**: Existing fine-tuning methods often treat visual and textual data separately, leading to a lack of integration between the two modalities. This isolation results in models that may excel in either visual understanding or textual comprehension but struggle to combine these insights effectively.

By conditioning image captioning on accompanying web content, **VisCon-100K** ensures the generated captions are **unique** and **contextually relevant** even as the dataset scales. This approach

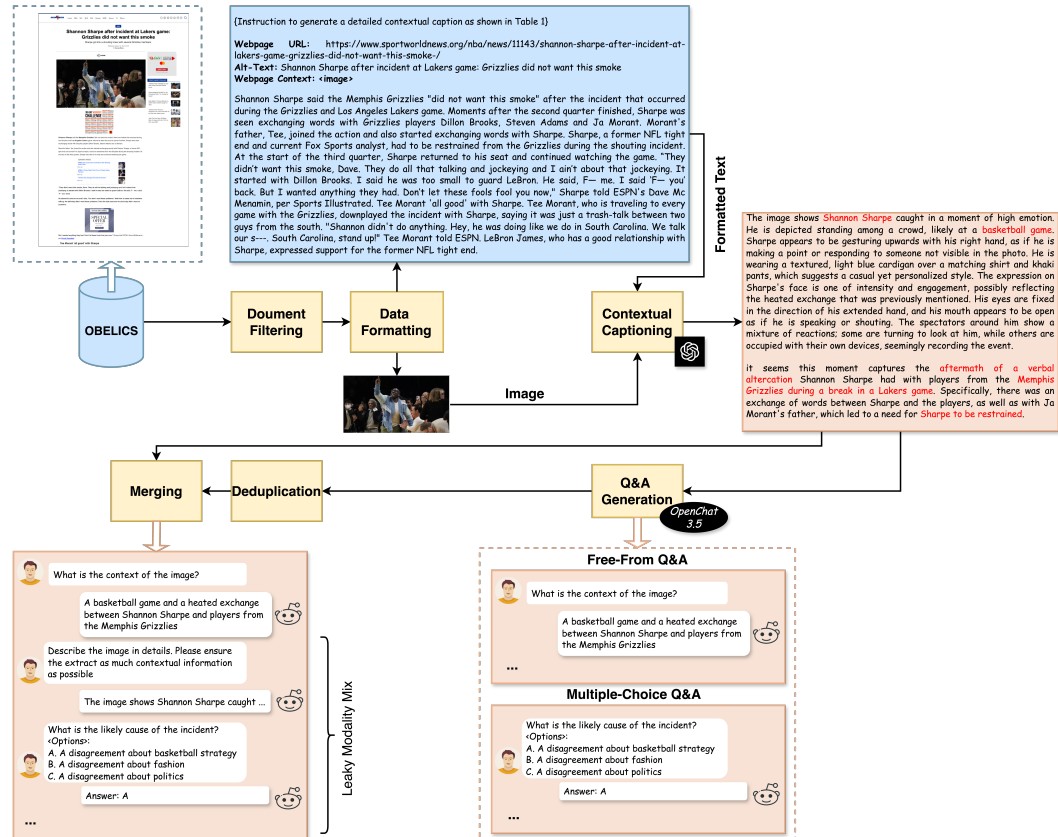

Figure 2: Data generation pipeline for creating the VisCon-100K dataset.

mitigates redundancy and enhances the dataset's relevance by leveraging the surrounding web context, thereby offering a more comprehensive training resource. Figure 1 illustrates this approach, showing a web page containing an image along with its non-contextual and contextual captions. The non-contextual caption describes the image in isolation, while our contextual caption integrates relevant information from the surrounding web content, providing a more nuanced and comprehensive description. Furthermore, our adaptation of the **leaky modality mix** in conversations provides an opportunity for interplay between visual and textual modalities with their tighter integration potentially.

## 3 DATA GENERATION PIPELINE

Our approach leverages interleaved image-text web documents to generate, **VisCon-100K**, a contextually rich fine-tuning dataset for vision-language models (VLMs). The data generation pipeline involves several steps: document filtering, contextual captioning, Q&A generation, deduplication and merging. The entire process is illustrated in Figure 2. We provide a detailed datasheet of VisCon-100K in Appendix A and show its properties along with example conversations in Appendix B.

### 3.1 DOCUMENT FILTERING

We begin by filtering the OBELICS web documents to include only those with a maximum of 2000 text tokens, as determined by the Vicuna-7b (Zheng et al., 2024) tokenizer. This step ensures that each document provides sufficient context while remaining manageable in size. Notably, more than 90% of the documents in OBELICS contain fewer than 2000 tokens.

## 3.2 CONTEXTUAL CAPTIONING

To generate contextual captions, we initially tested open-source VLMs like ShareGPT4V and LLaVA v1.5. However, we found that these models were not fine-tuned with web-contextual grounding datasets and often failed to include sufficient contextual information, sometimes even introducing hallucinations. In our qualitative evaluation with 100 samples, we discovered that GPT-4V significantly outperforms these models in producing high-quality contextual captions, especially when compared to non-contextual captions. Hence, we choose GPT4-V for this stage.

For each filtered web document, we extract relevant contextual information, including the webpage URL, image alt-text, and surrounding text. We also incorporate `<image>` and `<another-image>` placeholders to indicate the locations of the primary image and other images within the text. These elements collectively enhance the grounding of the captions, providing a rich context that helps in generating more fine-grained, accurate, and informative descriptions. Our approach was qualitatively validated, confirming its effectiveness. The prompt we adopted in using GPT-4V for generating contextual captions is shown in Table 2 in the Appendix.

## 3.3 Q & A GENERATION

Following the generation of contextual captions, we explored various large language models (LLMs) for creating diverse free-form and multiple-choice question-answer pairs. After experimenting with LLaMA2-7b (Touvron et al., 2023) , Mistral (Jiang et al., 2023), Vicuna-7b (Zheng et al., 2024), OpenChat 3.5 (Wang et al., 2023), and Gemma-7b (Team et al., 2024) on 100 samples, we qualitatively chose OpenChat 3.5, a 7-billion-parameter LLM, for its superior performance in Q&A generation. For the Q&A conversion, we found that open-source model like OpenChat 3.5 was sufficiently effective without the need to experiment with GPT-4V.

The Q&A generation is guided by a prompt adapted from LLaVA (Liu et al., 2024b) to convert captions into conversations, including few-shot examples for generating free-form question answers. We modified the instructions and few-shot examples also to generate multiple-choice questions. These prompts are shown in Tables 3 and 4 in the Appendix. Additionally, we implemented post-processing steps, such as matching identifier names with regular expressions and checking for pairs, to filter out poorly formatted outputs.

Including Q&A pairs is essential, especially when scaling the dataset. At 100K samples, VisCon-100K constitutes roughly 15% of the overall fine-tuning data. As we scale beyond 1 million samples—given our source dataset OBELICS has 353 million images—the percentage of VisCon will be much higher. In such a scenario, the role of Q&A becomes more crucial, as it reduces the model's bias towards always generating detailed responses irrespective of the question asked.

## 3.4 DEDUPLICATION AND MERGING

We merge the generated contextual captions, free-form, and multiple-choice question-answer pairs into coherent image conversations. Since captions do not inherently have an input prompt, we create a question for each caption using a randomly chosen LLaVA prompt for detailed image description and add the extra instruction `"Please ensure to extract and provide as much contextual information as possible."`

Given the observed duplication between free-form and multiple-choice questions, we perform deduplication to avoid redundancy and ensure a balanced representation of question types. The deduplication process involves the following steps:

- **Generate Sentence Embeddings**: Encode the questions into embeddings using AnglE model (Li & Li, 2023) to compute the cosine similarity matrix.
- **Select Unique Questions**: Iteratively select the most unique questions while maintaining a minimum count for each Q&A type (free-form and multiple-choice) using similarity scores.
- **Shuffle Conversation Rounds**: Shuffle the conversation rounds to avoid pattern bias in the order of questions and answers.

We include both captions and Q&A pairs in each dataset sample, despite potential overlaps in information. We term this approach as a **'leaky modality mix'**. This method integrates questions that

can be answered from both the image and the contextual caption within a single conversation sample, creating a controlled overlap or "leakage" of information across modalities. Our experiments in Section 5.3 show that this leaky modality mix performs better than non-leaky combinations of captions and Q&A pairs.

## 4 CONTEXTUAL CAPTIONING MODEL

To facilitate further extensions and reduce reliance on the paid GPT-4V service, we trained a contextual captioning model using the 100K contextual captions generated in our dataset. We fine-tuned IDEFICS2-8b, to accept both images and web content as input, enabling them to produce contextual captions. This additional fine-tuning with our dataset ensures that these models can generate high-quality contextual captions without the need for GPT-4V.

## 5 EXPERIMENTS

To evaluate the effectiveness of **VisCon-100K**, we conducted comprehensive experiments using two state-of-the-art vision-language models: ShareGPT4V-7b and IDEFICS2-8b. Our goal was to assess the impact of integrating VisCon-100K into existing fine-tuning datasets and to explore the performance benefits of the "leaky modality mix."

We did not directly compare our dataset with other VQA datasets because VisCon-100K is designed to complement, not replace, existing datasets. Importantly, while most other datasets focus on detailed image descriptions, our dataset includes contextual knowledge that extends beyond the image but remains closely related. To the best of our knowledge, we are the first to incorporate large-scale contextual information into a VQA dataset for vision-language models.

Additionally, we evaluated our dataset against its non-contextual version derived from the same source with the same number of images. This approach aligns with methods used in other vision-language datasets like LLaVA and ShareGPT4V. This experimental comparison demonstrates the effectiveness of VisCon-100K, highlighting the value of adding contextual information to enhance performance in vision-language tasks.

### 5.1 SETUP

For our experiments, we used the following setup:

- **Models**: We utilized the pre-trained versions of ShareGPT4V-7b (Chen et al., 2023) and IDEFICS2-8b (Laurençon et al., 2024b). For ShareGPT4V-7b, we performed full fine-tuning, while for IDEFICS2-8b, we employed parameter-efficient fine-tuning as recommended. Notably, for IDEFICS2-8b, we omitted image splitting, focusing instead on demonstrating the effectiveness of our data pipeline rather than optimizing for peak performance. Except for this, we followed the hyperparameters used in their original papers.

- **Fine-Tuning Data**: The fine-tuning setup for these models followed similar procedures as outlined in their original works, using their respective publicly available fine-tuning datasets. We augmented these datasets with 100K samples from VisCon-100K, roughly constituting a 15% increase in data volume.

- **Training Infrastructure**: We finetuned the models using AWS SageMaker instance of type ml.p4d.24xlarge, equipped with 8×40 GB A100 GPUs. This took a maximum of 12 hours for 1 epoch.

- **Framework**: Both models were trained using Hugging Face Transformers with DeepSpeed for optimization.

### 5.2 EVALUATION BENCHMARKS

We assessed model performance across six diverse vision-language benchmarks:

Table 1: Performance of ShareGPT4V-7b model for different configurations on the SEED benchmark.

| Configuration | Score |
|---|---|
| Base Model (without contextual data) | 66.24 |
| *Isolated Variants:* | |
| + Contextual Captions Alone | 66.9 |
| + Free-form Q&A Alone | 65.26 |
| + Multiple-choice Q&A Alone | 63.97 |
| *Non-Leaky Mix Variants:* | |
| + Combination of Free-form and Multiple-choice Q&A | 61.25 |
| + Separated Samples of Captions and Q&A | 59.31 |
| *Leaky Modality Mix:* | |
| + Combined Mix of Captions and Q&A | **67.62** |

- **SEED-Image** (Li et al., 2023): Comprising 14,232 samples, this benchmark covers categories like instance attributes, identity, interaction, location, counting, scene understanding, spatial relations, text understanding, and visual reasoning.

- **MMBench** (Liu et al., 2023b): With 6,666 samples, it includes perception and reasoning subcategories, such as coarse and fine-grained perception and relational, attribute, and logical reasoning.

- **MMMU** (Yue et al., 2024): Featuring 11,500 samples from fields like accounting, biology, chemistry, engineering, literature, medicine, physics, psychology, and more.

- **AI2D** (Kembhavi et al., 2016): Includes 5,000 images with three questions per image, covering various academic topics.

- **ScienceQA** (Lu et al., 2022): Consists of 2,000 samples across topics like astronomy, biology, geography, history, and physics.

- **LLaVA Bench** (Liu et al., 2024b): Contains 24 images with 60 questions focusing on visual conversation, detailed image descriptions, and complex visual reasoning. For scoring the answers, we used LLaMA3-8b for cost efficiency instead of GPT-4, comparing generated answers to reference texts.

### 5.3 Evaluating Data Combinations: The Impact of Leaky Modality Mix

To determine the optimal data composition, we evaluated different configurations of VisCon-100K using the SEED benchmark with the ShareGPT4V-7b model. This step was crucial to identify the best approach for integrating captions and Q&A pairs. We experimented with the following configurations:

- **Contextual Captions Alone**: Using only the contextual captions.

- **Free-form Q&A Alone**: Incorporating only the derived free-form question-answer pairs.

- **Multiple-choice Q&A Alone**: Using only the multiple-choice question-answer pairs.

- **Combination of Free-form and Multiple-choice Q&A**: Integrating both types of Q&A pairs in each conversation but no captions.

- **Separated Samples**: Using one conversation sample for captions and another for Q&A pairs.

- **Combined Mix**: Incorporating a mix of all three (contextual captions, free-form Q&A, and multiple-choice Q&A) in each sample.

The performance for each configuration is shown in Table 1. Our results show that the **'leaky modality mix'**—a configuration where each sample includes questions that can be answered from both the image and its contextual caption—outperforms using captions or Q&A pairs exclusively. This mix

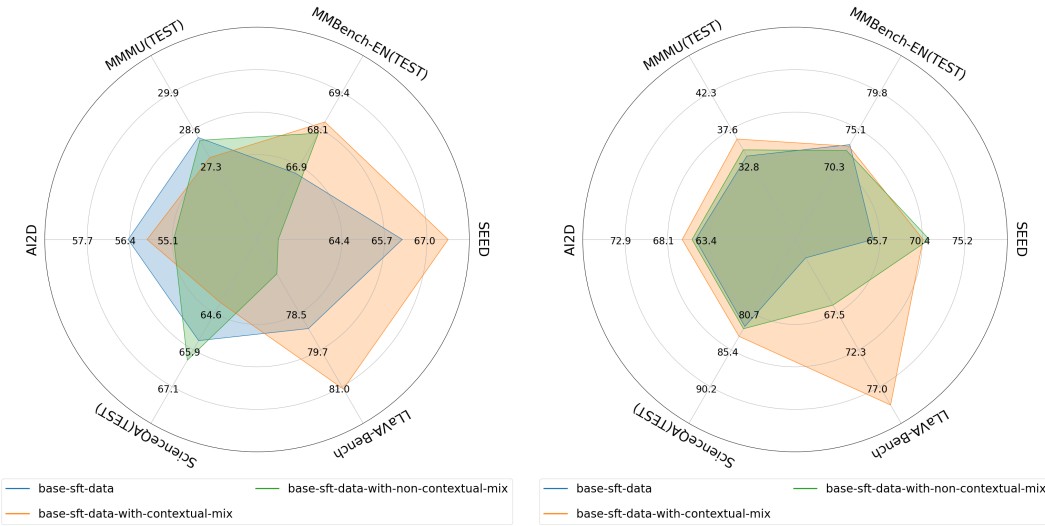

Figure 3: Performance of ShareGPT4V-7b model across 6 benchmarks for different data configurations

Figure 4: Performance of IDEFICS2-8b model across 6 benchmarks for different data configurations

mitigates biases seen in configurations using only captions (which tend to generate lengthy descriptions) or only Q&A pairs (which can overlook significant details). Additionally, by including both sources of information within a single conversation, the model can leverage the interplay between visual and textual data more effectively, leading to better integration and improved performance.

We also tested non-leaky mix configurations where captions were removed entirely or where captions and Q&A pairs were split into different samples, to understand the impact of explicit information leakage. The findings indicate that controlled leakage across modalities enhances the model's ability to integrate visual and textual information, thereby improving overall performance. Although the improvement from the leaky modality mix over using contextual captions alone appears modest, statistical tests confirm its significance. McNemar's test between the leaky modality mix and the base model yields a p-value of $2.118 \times 10^{-5}$, and between the leaky modality mix and the contextual captions model, a p-value of $0.027$—both indicating strong statistical significance.

### 5.4 Contextual vs. Non-Contextual Data

To get the non-contextual data, we followed the same pipeline described in Section 3, but without incorporating the webpage context during captioning and using prompts adapted accordingly.

Using the optimal **leaky modality mix**, we extended our evaluation across all six benchmarks. The results, depicted in Figure 3, demonstrate that the contextual mix outperforms in 3 out of 6 benchmarks. Specifically, the contextual mix significantly boosts performance on SEED and LLaVA Bench. On average, across all benchmarks, the contextual mix scored the highest with an **average of 60.81**, followed by the base model at 60.35, and the non-contextual mix at 59.51.

Interestingly, adding non-contextual data did not provide any substantial benefit on average, likely due to redundancy with the base fine-tuning data. This observation suggests that contextual information is crucial for enhancing the dataset's utility in vision-language tasks.

### 5.5 Generalizability

To assess the generalizability of our findings, we replicated the experiments with the more recent IDEFICS2-8b model, which is pretrained using interleaved image-text web documents, including OBELICS. Despite deriving our additional fine-tuning data from OBELICS itself, the contextual mix further improved IDEFICS2-8b's performance, as shown in Figure 4. In detail, across all benchmarks, the contextual mix scored an **average of 68.21**, compared to 65.50 for the non-contextual mix

and 63.31 for the base model. In addition, the boost across different benchmarks is more consistent compared to the experiments with ShareGPT4V-7b. We attribute this to the stronger integration of image and text data provided by our contextual conversations.

Preliminary experiments were conducted with ShareGPT4V-7b as IDEFICS2-8b was not available during the initial stages of our research. Given the significant computational resources required for fine-tuning and evaluation, we focused subsequent tests on the most promising configurations. The consistent performance improvements with IDEFICS2-8b underscore the utility of VisCon-100K, suggesting potential for further enhancements by processing additional web documents at scale.

It is crucial to note that the performance improvement is not solely due to the increased dataset size but also due to our approach. Table 1 shows that adding the same number of samples in non-leaky or isolated variants resulted in poorer performance. Figures 3 and 4 further illustrate that adding contextual samples yields better results than adding non-contextual counterparts across multiple benchmarks, highlighting the impact of contextual information.

## 5.6 CONTEXTUAL CAPTIONER

To facilitate further extensions of VisCon-100K, we finetuned IDEFICS2-8b model using the 100K contextual captions in our dataset. Evaluations on a held-out set of 1894 GPT-4 generated contextual captions showed an **increase of 4 BLEU points** and **3 ROUGE-L F1 points** with finetuning.

## 6 CONCLUSION

In this work, we introduced **VisCon-100K**, a novel dataset derived from interleaved image-text web documents, designed to enhance the fine-tuning of vision-language models (VLMs). Our approach generates contextually rich image conversations by creating image-contextual captions and transforming them into diverse question-answer pairs. Experiments demonstrate that integrating VisCon-100K notably improves VLM performance across multiple benchmarks. Additionally, our **leaky modality mix** strategy enhances the interplay between visual and textual modalities. We also provide a contextual captioner to facilitate the scalable extension of VisCon-100K, supporting open-source research and applications.

## 7 FUTURE WORK

1. **Multilingual Contexts and Scaling**: Extend the dataset to include multilingual web content, improving the generalizability and applicability of VLMs across different languages and cultural contexts. Additionally, scale the dataset to potentially over 300 million images, leveraging the full scope of the OBELICS dataset to enhance the depth and diversity of the fine-tuning data.

2. **Expanding Data Types for Fine-tuning**: Incorporate more complex conversation types, such as dialogues involving multiple images or more intricate Q&A formats, supported by ablation studies to determine the optimal mix of data types.

3. **Advanced Post-Processing Techniques**: Develop sophisticated post-processing methods to ensure the uniqueness, harmlessness, and usefulness of the generated data, enhancing the dataset's reliability and safety.

4. **Creating Diverse Benchmarks**: Establish comprehensive benchmarks to evaluate models on contextual visual question answering tasks, ensuring robust and generalizable model performance across varied scenarios.

## 8 LIMITATIONS

Despite the promising results, our approach has some limitations:

1. **Potentially Harmful Content**: While web data offers diverse contexts, it may include harmful or inappropriate content that our current pipeline does not explicitly filter out.

Future work should incorporate robust content moderation techniques to mitigate these risks.

2. **Reliance on GPT-4**: The use of GPT-4 for generating seed contextual captions provides a high-quality foundation for our dataset. However, GPT-4's performance in non-English languages and its reliance as a paid service may limit accessibility and introduce language biases. Our contextual captioner partially aims to address this by providing an open-source alternative, but further refinement is needed for broader applicability in multiple languages.

3. **Quality of Contextual Information**: The quality and relevance of the contextual information extracted from web documents can vary significantly, potentially affecting the consistency and effectiveness of the fine-tuning data. Ensuring high-quality context extraction remains a challenge that requires continuous improvement.

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

# A  VISCON-100K DATASHEET

## A.1  MOTIVATION FOR DATASET CREATION

- **Why was the dataset created?** To address the scarcity of high-quality, diverse visual fine-tuning data for VLMs, specifically focusing on contextual richness and real-world diversity.
- **Other potential uses:** Suitable for various vision-language tasks such as visual question answering, image captioning, conversational AI, and research on integrating visual and textual modalities.
- **Has the dataset been used?** Yes, initial results are presented in this paper (Section 5).
- **Funding:** Not answered to maintain anonymity.

## A.2  DATASET COMPOSITION

- **Instances:** Conversations consisting of images, contextual captions, free-form Q&A pairs, and multiple-choice Q&A pairs, linked to source web documents in OBELICS.
- **Relationships:** No explicit relationships between instances.
- **Quantity:** 100,000 image conversation samples.
- **Data per instance:** Images, contextual captions, free-form and multiple-choice Q&A pairs.
- **Reliance on external resources:** Relies on web documents from OBELICS (Laurençon et al., 2024a).
- **Recommended splits/evaluation:** Use standard VQA and captioning benchmarks (see Section 5).
- **Initial experiments:** Refer to Section 5.3 for the initial experiments conducted on the dataset.

## A.3  DATA COLLECTION PROCESS

- **Collection method:** Sampled from OBELICS, with captions generated using GPT-4V and Q&A pairs generated using OpenChat 3.5 (see Section 3).
- **Participants:** Automated processes; no human participants.
- **Dataset Time-frame:** Matches the web crawling timeframe of OBELICS (Laurençon et al., 2024a).
- **Data acquisition:** Derived from processed web documents in OBELICS.
- **Completeness:** Sampled 100,000 images due to cost and compute constraints.
- **Population:** OBELICS has 353 million images, potentially expandable with further crawling.
- **Missing data:** Documents with over 2000 tokens were excluded.

## A.4  DATA PREPROCESSING

- **Preprocessing:** Refer to Section 3.
- **Raw data saved:** Raw web documents along with images are retained.
- **Preprocessing software:** We utilize open-source python packages in our codebase. We plan to release the codebase.
- **Motivation achievement:** Yes, see Section 5.

## A.5  DATASET DISTRIBUTION

- **Data Distribution:** Refer to Section B.
- **Release date:** To be announced.
- **License:** Permission secured; exact licensing terms to be decided soon.
- **Fees:** None.

## A.6 DATASET MAINTENANCE

- **Supporting/hosting/maintaining the dataset:** Not answered to maintain anonymity.

- **Contact:** Not answered to maintain anonymity.

- **Updates:** We plan to extend the dataset based on directions mentioned in Section 7.

- **Repository:** We plan to create a public GitHub repository with a link to our dataset and documentation.

- **Documentation and communication of updates/revisions:** We plan to update them in a public GitHub repository.

- **Extensions/augmentations:** Refer to Section 7.

## A.7 LEGAL AND ETHICAL CONSIDERATIONS

- **Informed consent:** N/A

- **Ethically protected subjects:** N/A

- **Ethical review:** N/A

- **Consent for use:** N/A

- **Fairness considerations:** Our base dataset OBELICS (Laurençon et al., 2024a) incorporates ethical principles and content filters to minimize biases, but it inherits ethical concerns typical of large web-crawled datasets, such as unintended biases and under-representation of certain demographics. This may reflect in VisCon-100K as well, and ethical evaluations will be considered for future releases to address potential biases.

- **Sensitive information:** No obvious Personally Identifiable Information (PII) texts were found in our base dataset OBELICS (Laurençon et al., 2024a), except for public personalities and people mentioned in news articles.

- **Potential harm:** Possible exposure to harmful content due to the nature of web data. Mitigation efforts are planned for future work.

- **Inappropriate content:** The quality and appropriateness of content derived from diverse web sources can vary significantly, potentially leading to the inclusion of material that may be deemed inappropriate or offensive. Ensuring consistent content filtering and moderation remains a challenge that requires continuous enhancement to maintain the dataset's reliability and suitability for all audiences.

## B VISCON-100K PROPERTIES

This section presents various data properties of the VisCon-100K dataset.

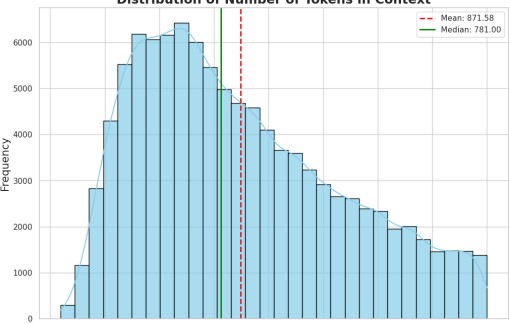
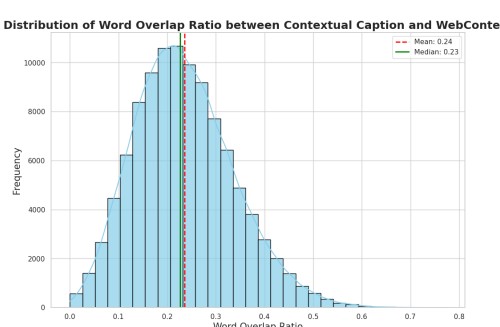

(a) Distribution of Number of Tokens in the Source Context.

(b) Distribution of Word Overlap Ratio between Contextual Caption and Source Context

Figure 5: Textual Characteristics of Source Context and their transformed Contextual Captions

In Figure 5a, the histogram illustrates that most web documents have a token count between 500 and 1000, indicating a substantial amount of context for generating rich image captions. The mean and median values suggest a slightly skewed distribution, with a long tail extending towards higher token counts.

Figure 5b shows the distribution of overlap ratio between contextual caption and the source context which is calculated after removing stopwords and stemming, and normalized by caption length. The average overlap ratio of 0.24 demonstrates the utility of VisCon-100K in augmenting image descriptions with relevant contextual information.

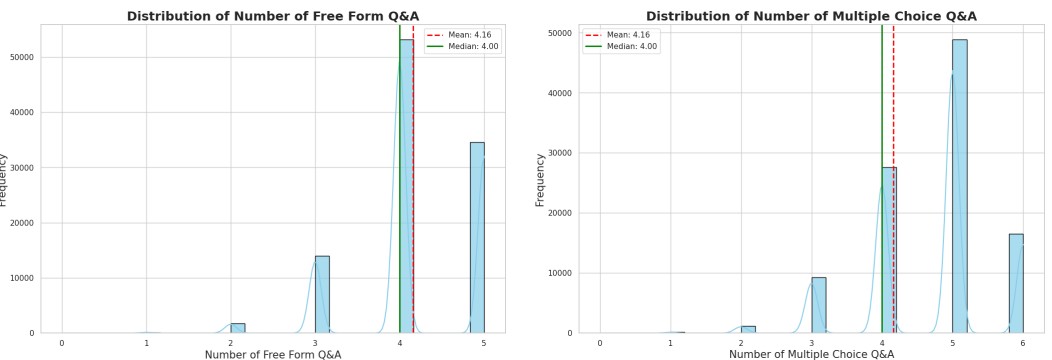

(a) Distribution of Number of Free Form Q&A     (b) Distribution of Number of Multiple Choice Q&A

Figure 6: Distributions of Q&A Types

The plot in Figure 6a shows that the majority of samples contain 4 free-form Q&A pairs, which aligns with the dataset's design to provide detailed conversational data.

Figure 6b illustrates most samples also contain 4 multiple-choice Q&A pairs. The similar distribution patterns between free-form and multiple-choice Q&A pairs facilitate a balanced training approach, allowing models to handle both types of queries effectively.

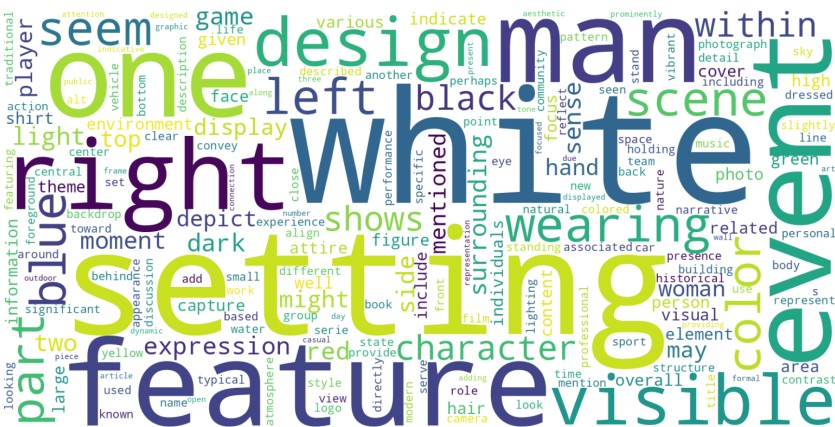

Figure 7: Word Cloud of Captions.

Figure 7 highlights frequently occurring terms such as "one," "white," "right," and "scene," reflecting the common descriptive elements in the dataset's image captions. The prominence of specific terms suggests a focus on detailed visual descriptions, which is critical for enhancing visual understanding in VLMs.

Few examples of the VisCon-100K dataset in Figure 8 demonstrate how contextual information from the web pages is used to enhance image descriptions and Q&A pairs, providing a comprehensive understanding of each image.

**Q1) What do you think is going on in this snapshot? Please ensure to extract and provide as much contextual information as possible.**
The image shows a first-person view from the video game Wolfenstein: Youngblood, where the player appears to have discovered a new fast-travel point—a subway station—within the game world. The central focus is a large, interactive metro map on a wall, similar to those found in real-world subway systems. The map is detailed with lines and various text reading "METRO UNLOCKED!" suggests that the player has just unlocked this station for travel. To the right of the map, there's a partial view of the player's firearm, a scoped rifle, indicating the game's action element. The image conveys an in-game moment where the player is being notified that they can now use this point to quickly travel between districts of the virtual Paris...
...
**Q4) What is the game Wolfenstein: Youngblood? <Options>: A. A first-person shooter game** B. A role-playing game C. A strategy game D. A puzzle game
A
**Q5) How many different elements can you see on the HUD of the player in the image?**
The HUD of the player in the image displays several elements. There's a mini-radar on the top right corner, several numbered icons indicating skills or equipment at the bottom right, and player health and armor indicators at the bottom left. These HUD elements are typical in first-person shooter games and provide the player with important information at a glance.
...

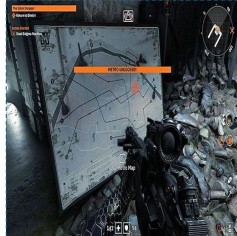

**Webpage URL:** https://guides.gamepressure.com/wolfenstein-youngblood/guide.asp?ID=50898
**Alt Text:** As you progress with the game, you can discover new subway stations (example is shown in the picture) - How I can use the fast travel option in Wolfenstein Youngblood? - FAQ - Frequently Asked Questions - Wolfenstein Youngblood Guide
**Webpage Context:** Wolfenstein: Youngblood has a fast-travel mechanism. On this page we explain how to use it - sisters can travel between districts of Paris and quickly return to the hideout of the resistance.
<another-image>It is the most obvious and the only way to travel between different districts of Paris - various locations are not physically connected to each other. As a result, underground tunnels are the only way to reach other parts of the city.
This mechanism can be tested by you for the first time in the final part of Riverside mission when you reach the subway station.
<another-image>When you check the map, a list of known stations will be displayed. These are the main districts of the city and the Catacombs also known as hiding place of the local resistance.
<image>
As you progress with the game, you can discover new subway stations (example is shown in the picture). From now on, they will be displayed in the list of available points of interest for fast-travel.
<another-image>
...

**Q1) What is the color of the pillow in the image? <Options>: A. Green B. Purple C. Yellow D. Blue**
B
**Q2) Can you tell me how many characters are visible in this image?**
In the image, we can see three characters. There is Luigi, easily identified by his green hat and distinctive mustache, lying down and seemingly asleep on a purple pillow. Above Luigi's head, a star hovers, possibly representing an item or an aspect of gameplay. To his side, we can also observe a charming yellow character with a resemblance to a round, happy star or perhaps a creature from the game's universe, which could be a friend or guide within Luigi's dreams.
**Q3) What is the character doing in the image? <Options>: A. Jumping B. Sleeping C. Running D. Eating**
B
**Q4) What is the character in the image likely to be? <Options>: A. A friend B. A guide C. An enemy D. A pet**
B
**Q5) Describe the following image. Please ensure to extract and provide as much contextual information as possible.**
The image depicts a close-up view of a Nintendo 3DS gaming system screen displaying a scene from the video game "Mario & Luigi: Dream Team". On the screen, we can see the character Luigi, easily identified by his green hat and distinctive mustache, lying down and seemingly asleep on a purple pillow. His eyes are closed, and he displays a peaceful expression. Above Luigi's head, a star hovers, possibly representing an item or an aspect of gameplay.
...

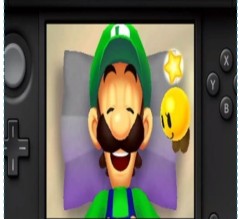

**Webpage URL:** https://brutalgamer.com/2013/04/17/nintendo-direct-round-up-lots-of-3ds-news/
**Webpage Context:** Home / 3DS / Nintendo Direct Round Up: Lots Of 3DS News
Nintendo Direct Round Up: Lots Of 3DS News
The latest Nintendo Direct gives us a lot to chew on.
<image>
A new Nintendo Direct broadcast aired today, and brought a bunch of news primarily aimed at the 3DS, with new details on existing games and brand new games announced.
First off, the first 3DS iteration of the Mario & Luigi RPG series, entitled 'Dream Team', is set to come out in North American on July 11th and Europe on July 12th, with the game based in Luigi's dreams.
A new Zelda game has been announced for the 3DS as well, and will be set as a sequel to the excellent SNES game, 'A Link To The Past'. It will include the ability for Link to become a drawing and cling to walls, and will use the 3D capabilities of the system. No name or release was announced for this.
In other news, Shin Megami Tensai 4 will be released in North America on July 16th, Game & Wario will be out in North America on June 23rd and Professor Layton and the Azran Legacy will reach the EU later this year and America some time in 2014.
...

**Q1) What is the main difference between the before and after images? <Options>: A. The house color changed B. The deck is removed C. The under-deck space is transformed into a covered patio area D. The deck is extended**
C
**Q2) What is the purpose of the new roof under the deck? <Options>: A. To provide shelter from the elements B. To increase the value of the house C. To provide additional storage space D. To create a new outdoor room**
A
**Q3) Write a detailed description of the given image. Please ensure to extract and provide as much contextual information as possible.**
The image presents a before-and-after comparison of a home improvement project focusing on a deck and the space beneath it. On the left side of the image, labeled "Before," we see a two-story house with light beige siding and white trims. There is a raised deck on the second level with spindled railings, accessible by a door. Below the deck, there is a visible under-deck space that seems unfinished and unused, with exposed decking joists and a sloped backyard with grass.
On the right side, labeled "After," the same view of the house shows significant changes. The under-deck space has been transformed into a covered patio area, with a new roof installed below the deck to provide shelter. This roof appears to be supported by sturdy white posts that match the house's trims and has a finished ceiling, which hides the underside of the deck and creates a more aesthetically pleasing and functional outdoor area. The posts are secured at the base with concrete footings, and the area beneath the deck now seems to be an inviting space suitable for outdoor furniture or recreational use, providing a shaded and protected area from the sun and rain.
...

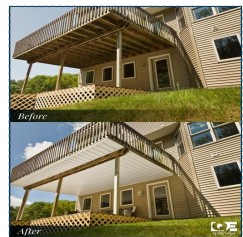

**Webpage URL:** https://housebeautifulus.netlify.app/how-to-build-a-deck-roof-youtube.html
**Alt Text:** Inside Out Under Deck Before And After Patio Under Decks Deck Remodel Under Decks
**Webpage Context:** The most common reason why people build roofs over decks is. Here are the top reasons why youll want a roof for yours. Frame up a box the size of the outer dimensions of your deck using 2 by 4 inch lumber.
<another-image><another-image><another-image><another-image><another-image><another-image><another-image>
Build decks on flat roofs with Bison Wood Tiles and Bison Deck Supports which is an adjustable pedestal for installing sophisticated modular style decks on.
How to build a deck roof youtube. Structural Design and Analysis including detailing. In this video i bring you along as we attempt to frame a roof on my parents deck.
...
Then run joists connecting the front of your roof with the rear every two feet along the length of your roof. I show you the difference between a flush beam and a dropped beam how. This is where the real planning began as the design of the roof would drastically effect the appearance of the finished deck.
<another-image><another-image><another-image><another-image><another-image><another-image> <image><another-image><another-image> <another-image><another-image><another-image><another-image><another-image>

Figure 8: Examples from the VisCon-100K dataset. The text, highlighted in red, shows contextual grounding.

864
865
866
867
868
869
870
871
872
873
874
875
876
877
878
879
880
881
882
883
884
885
886
887
888
889
890
891
892
893
894
895
896
897
898
899
900
901
902
903
904
905
906
907
908
909
910
911
912
913
914
915
916
917

Table 2: GPT-4 prompt template used to generate contextual captions for images.

```
Describe the image in detail.

Additionally, use the webpage's contextual information along
with the alt-text provided below to enrich the description.
Understand the webpage information based on its domain name.
Focus on the text surrounding the <image> tag, which denotes
the input image, and consider other images mentioned as
<another-image>.  Use only the webpage information relevant
to the input image and strictly ignore any information that is
not present in the input image.  Strictly do not mention the
webpage source in the description.

Webpage URL: {webpage_url}
Alt-Text:  {alt_text}
Webpage Context:  {webpage_context}
```

Table 3: Prompt template used to convert contextual captions to free-form Q&A pairs.

```
### Human:

You are an AI visual assistant, and you are seeing a single
image.  You are provided with the detailed description of the
same image you are looking at.  Answer all questions as you
are seeing the image.

Design a conversation between you and a person asking about
this photo.  Strictly use '<Human>' and '<Assistant>' as
identifiers and the conversation must have only 3 to 5 rounds.
The answers should be in a tone that a visual AI assistant
is seeing the image and answering the question.  Ask diverse
questions and give corresponding answers.

Include questions asking about the visual and the contextual
content found in the image description.  The visual content
covers the object types, counting the objects, object actions,
object locations, relative positions between objects, etc.
Only include questions that have definite answers:  (1) one
can see the content in the image that the question asks about
and can answer confidently; (2) one can determine confidently
from the image that it is not in the image.  Do not ask any
question that cannot be answered confidently.

Also include complex questions that are relevant to the
content in the image, for example, asking about background
knowledge of the objects in the image, asking to discuss about
events happening in the image, etc.  Again, do not ask about
uncertain details.  Provide detailed answers when answering
complex questions.  For example, give detailed examples
or reasoning steps to make the content more convincing and
well-organized.  You can include multiple paragraphs if
necessary.

Remember to not output more than 5 rounds.

{few_shot_examples}

Image Description:

{text}

### Assistant:
```

Table 4: Prompt template used to convert contextual captions to multiple-choice Q&A pairs.

```
### Human:

You are an AI visual assistant, and you are seeing a single
image.  You are provided with the detailed description of the
same image you are looking at.  Answer all questions as you
are seeing the image.

Design a set of multiple choice questions between you and a
person asking about this photo.  Strictly provide 4 choices
A., B., C. and D. where only one is valid.  and Strictly use
`<Human>`,`<Options>` and `<Assistant>` as identifiers for
the question, options (`` delimited) and answer (include only
the letter option), and the conversation must have only 3 to
5 rounds.  The answers should be in a tone that a visual AI
assistant is seeing the image and answering the question.  Ask
diverse contextual questions and give corresponding answers.
Additionally, questions should be independent from each
others.

Include questions asking about the visual and the contextual
content found in the image description.  The visual content
covers the object types, counting the objects, object actions,
object locations, relative positions between objects, etc.
Only include questions that have definite answers:  (1) one
can see the content in the image that the question asks about
and can answer confidently; (2) one can determine confidently
from the image that it is not in the image.  Do not ask any
question that cannot be answered confidently.

Also include complex questions that are relevant to the
content in the image, for example, asking about background
knowledge of the objects in the image, asking to discuss about
events happening in the image, etc.  Again, do not ask about
uncertain details.  Provide detailed answers when answering
complex questions.  For example, give detailed examples
or reasoning steps to make the content more convincing and
well-organized.  You can include multiple paragraphs if
necessary.

Remember to not output more than 5 rounds.

{few_shot_examples}

Image Description:

{text}

### Assistant:
```