# OpenReview forum: "VisCon-100K: Leveraging Contextual Web Data for Fine-tuning Vision Language Models with Leaky Visual Conversations"
_ICLR.cc/2025/Conference — ICLR 2025 Conference Withdrawn Submission_

### Official Review · Reviewer_YudP · 2024-10-20

**Soundness:** 3
**Presentation:** 3
**Contribution:** 2
**Rating:** 5
**Confidence:** 4

**Summary:**

This paper introduces a novel dataset VisCon-100K, designed to enhance vision-language models (VLMs). The dataset is constructed by using GPT-4V to generate image-contextual caption and later transformed to instruction data. The paper also introduces a 'leaky modality mix', where conversation samples contain questions answerable from both the image and its contextual caption.  The dataset shows strong performance with two popular VLM approaches, ShareGPT4V and IDEFICS2.

**Strengths:**

+ The paper introduces a visual instruction-tuning dataset with enhanced contextual information.
+ The paper demonstrates the effectiveness of the proposed 'leaky modality mix' data by extensive ablations.
+ The paper is well written and easy to follow.

**Weaknesses:**

+ **Lack of novelty**.
A very similar idea of merging the contextual information of an image with its description is already introduced in [1]. The paper lacks any citation or discussion about the difference between them.

+ **Unclear motivation and incomplete evaluation**.
I am somewhat confused by the effectiveness of the leaky modality mix. It seems counterintuitive that if the answer can be derived solely from the provided caption, the Vision Language Model (VLM) would learn a shortcut and disregard the image. This approach should not, in theory, be beneficial to the training process. A possible explanation for why this paper finds this type of data effective is that the model is primarily evaluated on benchmarks with significant text redundancies, where the image is not necessarily required to answer the question. Consequently, it is crucial to evaluate the model on benchmarks specifically designed to ensure visual reasoning, such as those presented in [2] and [3], to thoroughly validate the authors' conclusions.

+ **Limited improvement**.
The improvement in Fig. 3 and Fig. 4 seems marginal on several benchmarks. Training with the baseline data often surpasses training with the proposed dataset, which makes it uncertain whether the dataset truly brings improvement.


[1] Yu, Qiying, et al. "Capsfusion: Rethinking image-text data at scale." Proceedings of the IEEE/CVF Conference on Computer Vision and Pattern Recognition. 2024.

[2] Zhang, Renrui, et al. "Mathverse: Does your multi-modal llm truly see the diagrams in visual math problems?." arXiv preprint arXiv:2403.14624 (2024).

[3] Chen, Lin, et al. "Are We on the Right Way for Evaluating Large Vision-Language Models?." arXiv preprint arXiv:2403.20330 (2024).

**Questions:**

I would appreciate further clarification about why leaky data can provide an opportunity for interplay between visual and textual modalities with their tighter integration potentially', as demonstrated in L198-200. Why won't the model learn a shortcut of only attending to the text?

---

### Official Review · Reviewer_owzc · 2024-11-01

**Soundness:** 2
**Presentation:** 2
**Contribution:** 2
**Rating:** 5
**Confidence:** 5

**Summary:**

This paper presents VisCon-100K, a novel dataset aimed at enhancing vision-language model (VLM) performance by incorporating contextual information from web sources. The authors designed an automated pipeline to generate this data, utilizing GPT-4V for contextual captioning and OpenChat 3.5 for generating both free-form and multiple-choice Q&A pairs. They also introduce a "leaky modality mix" approach, allowing samples to contain questions answerable by both image and context, which has shown superior performance over non-leaky setups. Additionally, authors provide a high-quality contextual captioner.

**Strengths:**

1. This paper focuses on the usage of contextual web data for vision-language models.

2. The proposed data construction pipeline is scalable.

3. Experimental results demonstrate the effectiveness of the proposed VisCon-100K dataset on ShareGPT4V and IDEFICS2.

**Weaknesses:**

1. The experimental settings are outdated. The authors should consider fine-tuning recent Multimodal Large Language Models (MLLMs), such as MiniCPM-V, InternVL1.5, and LLaVA-Next. These models are trained on diverse domain data, and demonstrating the effectiveness of your dataset on these state-of-the-art models is crucial. Given that these models were introduced before May, the lack of validation on these strong models, with only weaker models tested, makes the effectiveness of the proposed dataset unconvincing.

2. There is a lack of quality analysis for the proposed dataset. Since all data is model-generated, it is likely that a considerable amount of noise is present.

3. The paper also introduces a captioner, but it lacks performance analysis. Improvements in BLEU and ROUGE scores only indicate that the model-generated captions are more similar to the ground truth, which does not necessarily mean higher quality. The authors should provide results from human evaluation or GPT-assisted evaluation to strengthen the claims.

**Questions:**

See weakness.

---

### Official Review · Reviewer_5ygV · 2024-11-04

**Soundness:** 2
**Presentation:** 2
**Contribution:** 1
**Rating:** 3
**Confidence:** 3

**Summary:**

The paper proposed VisCon-100K, which is a new vision-language dataset addressing the need for high-quality visual fine-tuning data. Built from 45K web documents in the OBELICS dataset, VisCon-100K includes 100K image conversation samples featuring diverse question-answer pairs and image-contextual captions generated with GPT-4V and OpenChat 3.5. This dataset boosts VLM performance by incorporating contextual web data, surpassing traditional fine-tuning on solely visual content. Key contributions include the release of VisCon-100K, a trained contextual captioner, and the concept of "leaky modality mix" for more effective integration of visual and textual data.

**Strengths:**

I believe how to construct interleaved image-text data is important for developing VLM. This paper has yield some preliminary studies along this direction.

Overall, the proposed data generation pipeline is composed of reasonable components, which results in a 100k dataset.

They have listed their experimental details in a clear way. Also, some ablation studies are included. One finding is they noticed using leaky modality mix, i.e. QA can be inferred from both image and text, align with my intuition.

**Weaknesses:**

- I am unsure if the current evidence could support the effectiveness of the proposed VisCon-100K. So far, I only located there is a finetune experiment in Section 5.6 which finetuned IDEFICS2-8b and show some improvements over the generated caption evaluation. For this specific experiment, can the author provide more numbers on other datasets, such as the datasets used in Figure 4? Besides, could the author please do more finetuned experiments on other VLMs?

- Besides the finetuned experiments, I do not find other evidence to prove the proposed VisCon-100K is high-quality as well as the annotated pipeline could yield high-quality data. Can the author do other verification experiments, such as hire human-annotators to check if the data really high-quality.

- One of the most important experiments in the paper is Table1, however, it is only conducted in the SEED benchmark. Could the author also apply those ablations to other datasets to check if the improvements are consistent?

**Questions:**

Please address all the concerns raised above.

---

### Official Review · Reviewer_odMZ · 2024-11-04

**Soundness:** 3
**Presentation:** 1
**Contribution:** 2
**Rating:** 3
**Confidence:** 3

**Summary:**

This paper introduces VisCon-100k, a dataset specifically designed for fine-tuning large visual language models. The article begins by analyzing the limitations in the construction of existing visual language model SFT datasets, proposes improvements, and based on these, creates VisCon-100k. A major highlight of VisCon-100k is the incorporation of contextual information from OBELICS sample texts during its construction. Through experiments, the paper verifies the superiority of VisCon-100k to some extent.

**Strengths:**

- This paper analyzes the limitations in the construction process of existing visual language model SFT datasets. For example, ShareGPT4V uses only images as input to "distill" GPT4V, thus it can only extract generic, generalized descriptions from the images, and fails to capture more specific, fine-grained information such as time, location, and characters. Another example is LLaVA, which relies on manually annotated captions or object detection information, using purely linguistic LLMs to construct multimodal data. This limits the diversity and richness of the data and increases the risk of generating erroneous information. In response to these issues, this paper attempts to provide a solution by inputting both the image and its related text into GPT4V, enabling it to generate captions that include contextual information, thereby overcoming the aforementioned limitations to some extent.
- The paper demonstrates the effectiveness of this approach through experiments: by providing contextual information during the construction process to GPT4V, the generated captions contain contextual information, and the resulting SFT data is shown to be more effective than data without contextual information, thus validating the design of this method.

**Weaknesses:**

- The fairness of the experimental design needs to be strengthened: I noticed that the authors mention in the text (line 291) that the purpose of VisCon-100k is not to replace the existing SFT datasets but to complement. Nevertheless, the experimental section should demonstrate the advantages of **complementing with VisCon-100k** compared to **complementing with other SFT datasets**.
- The authors point out in the abstract (line 14) and introduction (line 86) that the existing SFT datasets are not high quality and lack diversity; then mention in (line 96) that VisCon-100k includes **fine-grained visual descriptions and broader range of contextual information**. However, the paper lacks a direct comparison with existing SFT datasets, including qualitative illustrations, quantitative statistics (e.g., comparing the diversity of different datasets using appropriate metrics), and the direct experimental comparisons mentioned in the previous issues.
- Although the experimental results in Figure 3 and Figure 4 demonstrate the advantages of contextual data over non-contextual data, to better prove the diversity of contextual data, it is suggested to provide more comparative information. Consider the following methods:
    - Perform statistical comparisons of the entities in the QA texts from both datasets.
    - Project the QA texts of both datasets into the feature space of a pre-trained Sentence BERT and use t-SNE for visual analysis.
- The writing quality of this paper is poor, making it difficult to read and understand.

**Questions:**

See **Weaknesses**

---

### Note · Authors · 2024-11-25

I have read and agree with the venue's withdrawal policy on behalf of myself and my co-authors.